# Deep Learning with Quantitative Features of Magnetic Resonance Images to Predict Biochemical Recurrence of Radical Prostatectomy: A Multi-Center Study

**DOI:** 10.3390/cancers13123098

**Published:** 2021-06-21

**Authors:** Ye Yan, Lizhi Shao, Zhenyu Liu, Wei He, Guanyu Yang, Jiangang Liu, Haizhui Xia, Yuting Zhang, Huiying Chen, Cheng Liu, Min Lu, Lulin Ma, Kai Sun, Xuezhi Zhou, Xiongjun Ye, Lei Wang, Jie Tian, Jian Lu

**Affiliations:** 1Department of Urology, Peking University Third Hospital, Peking University, Beijing 100191, China; yanye@bjmu.edu.cn (Y.Y.); haizhuixia@bjmu.edu.cn (H.X.); zyt100582@163.com (Y.Z.); chengliu@bjmu.edu.cn (C.L.); malulin@medmail.com.cn (L.M.); 2CAS Key Laboratory of Molecular Imaging, Beijing Key Laboratory of Molecular Imaging, the State Key Laboratory of Management and Control for Complex Systems, Institute of Automation, Chinese Academy of Sciences, Beijing 100190, China; lz_shao@seu.edu.cn (L.S.); zhenyu.liu@ia.ac.cn (Z.L.); 15120478@bjtu.edu.cn (K.S.); zxzxidian@163.com (X.Z.); 3School of Computer Science and Engineering, Southeast University, Nanjing 210096, China; yang.list@seu.edu.cn; 4CAS Center for Excellence in Brain Science and Intelligence Technology, Institute of Automation, Chinese Academy of Sciences, Beijing 100190, China; 5School of Artificial Intelligence, University of Chinese Academy of Sciences, Beijing 100080, China; 6Department of Radiology, Peking University Third Hospital, Peking University, Beijing 100191, China; heweihw@126.com (W.H.); hychen@bjmu.edu.cn (H.C.); 7Beijing Advanced Innovation Center for Big Data-Based Precision Medicine, School of Engineering Medicine, Beihang University, Beijing 100191, China; jgliu@buaa.edu.cn; 8Key Laboratory of Big Data-Based Precision Medicine (Beihang University), Ministry of Industry and Information Technology of the People’s Republic of China, Beijing 100191, China; 9Department of Pathology, Peking University Third Hospital, Peking University, Beijing 100191, China; lumin@bjmu.edu.cn; 10Urology and Lithotripsy Center, Peking University People’s Hospital, Peking University, Beijing 100044, China; yexiongjun@bjmu.edu.cn; 11Department of Urology, Beijing Friendship Hospital, Capital Medical University, Beijing 100050, China; sclare@163.com

**Keywords:** prostate cancer, biochemical recurrence, survival prediction, deep learning, MRI

## Abstract

**Simple Summary:**

Biochemical recurrence after radical prostatectomy is vitally important for long-term oncological control and subsequent treatment of these patients. We applied radiomic technique to extract features from MR images of prostate cancer patients, and used deep learning algorithm to establish a predictive model for biochemical recurrence with high accuracy. The model was validated in 2 indepented cohorts with superior predictive value than traditional stratification systems. With the aid of this model, we are able to distinghuish patients with higher risk of developing biochemical recurrence at early stage, thus providing a window to initiate neoadjuvant or adjuvant therapies for prostate cancer patients.

**Abstract:**

Biochemical recurrence (BCR) occurs in up to 27% of patients after radical prostatectomy (RP) and often compromises oncologic survival. To determine whether imaging signatures on clinical prostate magnetic resonance imaging (MRI) could noninvasively characterize biochemical recurrence and optimize treatment. We retrospectively enrolled 485 patients underwent RP from 2010 to 2017 in three institutions. Quantitative and interpretable features were extracted from T2 delineated tumors. Deep learning-based survival analysis was then applied to develop the deep-radiomic signature (DRS-BCR). The model’s performance was further evaluated, in comparison with conventional clinical models. The model achieved C-index of 0.802 in both primary and validating cohorts, outweighed the CAPRA-S score (0.677), NCCN model (0.586) and Gleason grade group systems (0.583). With application analysis, DRS-BCR model can significantly reduce false-positive predictions, so that nearly one-third of patients could benefit from the model by avoiding overtreatments. The deep learning-based survival analysis assisted quantitative image features from MRI performed well in prediction for BCR and has significant potential in optimizing systemic neoadjuvant or adjuvant therapies for prostate cancer patients.

## 1. Introduction

Prostate cancer (PCa) is the most common cancer among men and the second leading cause of death for men worldwide [1]. Radical prostatectomy (RP) is one option of the multimodal approaches for organ-confined or locally advanced PCa. Biochemical recurrence (BCR) after RP is known to harbor more advanced or aggressive disease. The 10-year estimated BCR rate after RP has been reported to be up to 27% [2]. More than two-thirds of BCR cases develop in the first two years after surgery [3]. BCR is known to be a surrogate of local recurrence, distant metastasis, and cancer-specific death [4]. Therefore, early identification of patients who are predisposed to developing BCR could provide a window for early intervention with systemic or local therapies.

In fact, the natural history of biochemical relapse is heterogeneous. It is difficult to stratify patients according to conventional TNM system, several clinical predictive models were gradually developed for BCR prediction. Among which, the Cancer of the Prostate Risk Assessment (CAPRA/CAPRA-S) score, National Comprehensive Cancer Network (NCCN) model and Gleason grade group (GG-RP) system were three kinds of most widely adopted models [5]. According to these systems, patients were assigned into low-, intermediate- and high-risk groups, while patients in the middle group would face controversial prognosis and unclear therapeutic guidance. Besides, these models only incorporate pure clinical factors, neglecting more comprehensive information obtained from imaging and genetic data. 

MRI has been routinely performed as one component of the multi-modal diagnostic procedure of prostate cancer. MRI can overcome the biopsy bias in a non-invasive fashion and better highlight tumoral heterogeneity. Adoption of the PI-RADs 2.1 protocol led to a sensitivity of 79% in distinguishing benign lesions of malignancies [6]. However, PI-RADs system demonstrated limited potentials for oncologic prognoses such as BCR or overall survival estimation. Radiomics has emerged as an approach converting conventional images into high volume quantitative features [7]. The radiomics may not only analyze anatomical information of micro structural but also information representing the underlying pathophysiologic process, which may be correlated with oncological prognosis. Thus, radiomics has successfully introduced to discriminate benign and malignant lesions [8], predict aggressiveness [9] and estimate Gleason scores [10]. However, there were still few studies focusing on BCR prediction for patients after radical prostatectomy with relatively large amount datasets [11,12,13].

In this study, based on pathological ground truth registration annotations, we developed a novel model for BCR prediction, combing quantitative features of magnetic resonance imaging (MRI) and deep learning-based survival analysis. We then validated the model and calculated its incremental predictive value in comparison with several acknowledgeable nomograms in a multicenter dataset.

## 2. Materials and Methods

This study was approved by the institutional review board of the Peking University Third Hospital Medical Science Research ethics committee with a waiver of informed consent and compliant with the principles in the Declaration of Helsinki (S2019326).

### 2.1. Study Design

A total of 584 consecutive patients diagnosed with prostate cancer were included from three institutions in Beijing. All patients were enrolled with strict inclusion and exclusion criteria. The inclusion criteria were shown as follows: (a) primary prostate adenocarcinoma confirmed by radical prostatectomy; (b) locally or locally advanced disease according to the 8th edition of the AJCC Staging Criteria; (c) No neoadjuvant androgen deprivation treatment (ADT) before surgery; (d) had documented BCR or, (e) did not have BCR but followed over 3 years. BCR was defined as two consecutive total PSA readings >0.2 ng/mL after RP [14]; (f) Radical prostatectomy was performed. The exclusion criteria were as follows: (a) patients with neoadjuvant ADT before surgery; (b) patients with <3 years of follow-up data without BCR status; (c) patients with incomplete clinical information; (d) patients with distant metastasis; (e) patients with other pathology types or mixed pathology types; (f) patients with postoperative PSA nadir > 0.1 ng/mL within 3 months after RP. According to the criteria, 485 patients were enrolled in the final analysis (Table 1). Patients who were free from BCR were censored at the last follow-up.

The patients were divided into three cohorts, the primary cohort (PC: *n* = 368, from Peking University Third Hospital, Beijing, China) and other two external validation cohorts (VC1: *n* = 34, from Beijing Friendship Hospital; VC2: *n* = 83, from Peking University People’s Hospital). Conceptual workflow was demonstrated in Figure 1. The first step included data acquisition of images and tumor segmentation. Then, we extracted radiomic features from tumor area for tumor heterogeneity. Thirdly, feature evaluation based on rank is used to roughly remove features that are of little value in modeling based on statistical approaches. Finally, prognosis prediction of BCR was constructed and validated in our multi-center dataset. Comparison with guidelines was also supplemented. Details of patient recruitment and study design were shown in Appendix A.

### 2.2. MRI Protocol

All patients received 3T MRI examination before systematic transrectal needle biopsy. All MR images were obtained with T1W, T2W, DWI and Apparent diffusion coefficient (ADC) maps. Only DICOM data of T2WI were used for analysis in this study. Details of scanning parameters were shown in Appendix A.

### 2.3. Annotations

Pathological hematoxylin-eosin sections of each patient from radical prostatectomy specimens were scanned at 40× magnification, converting to computational pathological sections (NanoZoomer S360, HAMAMATSU, Hamamatsu City, Japan). First, one pathologist of urology expertise delineated the lesions that were responsible for diagnosis on each section. Second, this pathologist and one urological radiologist together recognized and delineated lesions on MRI, which were correlated to pathological whole-mount slides. The tumor regions of interest (ROIs) were delineated using ITK-SNAP software (www.itksnap.org, 27 April 2018). Detailed examples of annotations were demonstrated in Appendix A.

### 2.4. Radiomic Features of Mp-MRI Extraction

Standard normal distribution of image intensities was obtained through Z-score normalization. Subsequently, 702 quantitative features extracted from the T2WIs of individual patients were calculated to characterize intratumoral heterogeneity and complexity. Five common groups of features (first-order, shape, texture, wavelet and Laplacian of Gaussian filter) were extracted by using “Pyradiomics” (Version 2.1.1, https://github.com/Radiomics/pyradiomics, 14 April 2021) [15]. Details of the features were summarized in Appendix A.

### 2.5. Deep Learning-Based Survival Analysis for Signature Construction to BCR Survival

In this study, we developed a deep survival neural network for BCR prediction and constructed a signature named deep radiomic signature for BCR (DRS-BCR) to predict BCR probability. After radiomic feature extraction, we firstly evaluated all the radiomic features in the PC by univariate analysis and recorded the concordance (C-index) and *p*-value (*p*) between each feature and BCR. Features with predictive potential (C-index > 0.5 and *p* < 0.05) were selected from the features pool to eliminate redundant information. During the modeling phase, we used a new design of neural network structure to describe the correlation between image features and BCR-free survival, which contained a dense box to gather information of multi-level abstraction [16], and an auto-coding box to generate sparse features [17]. Three hidden layers with forty-eight neurons were used for the dense box. Two hidden layers of forty-eight neurons and a hidden layer with twenty-four neurons were used in the auto-coding box. The dropout and early stop strategy were employed in the training process to mitigate overfitting. The loss function of the model was deep survival loss [18], in which the optimizer was Adam [19], and the training batch was the total amount of data in the PC. Finally, the output of the neural network was the risk of BCR for a patient, namely DRS-BCR. 

To evaluate the interaction between clinical indicators and our proposed DRS-BCR, important clinical factors with prognosis power and DRS-BCR were add in a Cox proportional-hazards regression model (CPH) for BCR survival. Comparisons of DRS-BCR were evaluated in all cohorts, including combination model and clinical factors (Appendix A).

### 2.6. Statistics

We compared the patients with and without BCR using the t-test for continuous variables and the chi-square test or Fisher’s test for categorical variables. The C-index and 95% confidence interval (CI) were calculated, in order to evaluate the performance of the BCR-free survival model. The Kaplan–Meier (K–M) method and log-rank test were used to estimate BCR-free survival. The time-dependent analysis (including ROC curves, AUC, sensitivity, and specificity) was performed to evaluate accuracy of predicting BCR. The cutoff in the time-dependent analysis was selected using criteria based on Youden Index. The Wilcoxon signed rank test was used to compare the C-index distributions of different models. All packages were based on R software (version 3.1.0). A two-sided *p* value < 0.05 was considered significant.

## 3. Results

### 3.1. Patient Characteristics

A total of 485 patients were finally enrolled in the study, with a median age of 69.86 [95% CI, 62.79–76.93] years old. The median BCR-free survival was 57.7 months. There were 81 (22.0%), 9 (26.5%) and 23 (27.7%) patients with 3-year BCR in the PC, VC1 and VC2, respectively, which was balanced (*p* = 0.479), detail information is shown in Table 1. 

To assess the independent prognostic power of each clinical parameter, we grouped them as preoperative clinical parameters (PSA, cT stage, GG-NB, PPB) and postoperative parameters (GG-RP, surgical margin, extracapsular extension, seminal vesicle invasion) in a univariable cox regression model. The parameters were normalized by CAPRA (cT stage, GG-NB, PPB) or CAPRA-S (PSA, GG-RP, surgical margin, extracapsular extension, seminal vesicle) criteria. The results of the univariate analysis are shown in Appendix A. 

We integrated all significant clinical variables (*p* < 0.05), and grouped them by the acquisition time (pre-operative, post-operative, a combination of both pre- and post-operative), and built three clinical prognostic models for BCR prediction, respectively (Appendix A). We termed the clinical variables as clinical signature (CS). The combination of pre- and post-operative clinical variables (CS-combine) yielded C-index [95% CI] of 0.693 [0.634–0.752] in PC, 0.651 [0.471–0.831] in VC1 and 0.641 [0.529–0.753] in VC2, which is the highest among the three. 

### 3.2. Evaluation of Quantitative Features of Mp-MRI for BCR-Free Survival

The radiomic pool of 702 features was evaluated by univariate analysis of BCR-free survival. From which, 155 features with both concordance and significance were finally selected for modeling. Among these, 125 (80.6%) of them were in texture category, 22 (14.2%) were in first-order category, and 8 (5.16%) were in shapes related category (Appendix A). Features with the highest concordance and significance were the intensity of gray value (first-order) from the original imaging.

### 3.3. Assessment of DRS-BCR for Predicting BCR-Free Survival

The DRS-BCR was generated from a deep survival radiomic neural network (DSNN) to predict BCR-free survival, which yielded C-index of 0.802 [95% CI, 0.758–0.846] in PC, 0.811 [95% CI, 0.722–0.9] in VC1 and 0.794 [95% CI, 0.718–0.87] in VC2. Time-dependent ROC curves were then applied to evaluate 3-year and 5-year BCR detection rates (Figure 2). The DRS-BCR of 3-year BCR yielded AUCs of 0.84 in PC, 0.85 in the VC1, and 0.84 in the VC2, respectively (Figure 3). The DRS-BCR of 5-year BCR yielded AUCs of 0.83 in PC, 0.88 in the VC1, and 0.88 in the VC2, respectively (Appendix A). The HR of DRS-BCR was 1.705 [95% CI, 1.531–1.893] (*p* < 0.001) in univariate analysis.

Next, we performed K–M analysis of 3-year BCR-free survival in the PC, VC1, and VC2 (Figure 2a–c). The K–M curves of cohorts all revealed significant differences by log-rank test between groups of high and low risk (*p* < 0.001). The calibration curve of the radiomics model estimated the probability of 3-year BCR, which demonstrated good agreement in the primary cohorts (Figure 2d). The Hosmer–Lemeshow test yielded a non-significant statistic (*p* = 0.657), suggesting no departure from the perfect fit. Good performance was also observed for the probability of 3-year BCR in all validation cohorts (*p* = 0.419, *p* = 0.583) (Figure 2e,f). The decision curve showed relatively good performance for the radiomic signature as well (Figure 2g–i). The results of incorporating DRS-BCR with clinical variables were evaluated in Appendix A.

### 3.4. Comparison between DRS-BCR and Clinical Nomogram

We performed Cox proportional hazards regression analysis for the GG-RP, CAPRA-S, NCCN, and CAPRA models in our multicenter dataset. Performances of these four conventional popular models for BCR prediction were listed in Table 2. Statistically, the GG-RP, CAPRA-S, NCCN, and CAPRA all demonstrated discriminative power for BCR prediction (HR > 1 and *p* < 0.001). Significant differences of C-index between DRS-BCR/DRC-BCR and the other four clinical models were tested by the U-statistics-based C estimator, and the C-index of DRS-BCR/DRC-BCR significantly outperformed others in multicenter validation (*p* < 0.05). For the performance of discriminating BCR at 3-year, IDI and continuous NRI tests were applied to evaluate the significance of the incremental performance. The test results showed that DRS-BCR maintained better concordance of 3-year BCR than other clinical models in all datasets (*p* < 0.05) (Appendix A). The K–M curves were further utilized to compare the models for assigning patients into high-risk and low-risk of BCR-free survival, the log-rank test was used to evaluate the significant discrimination (*p* < 0.05) between high-risk and low-risk. Comparisons of K–M analysis and log-rank tests between DRS-BCR and other conventional models with multi cut-off groups were carried out (Appendix A). DRS-BCR achieved stronger discriminative power than any other clinical models at almost any cut-off settings (*p* < 0.001). More comparisons between DRS-BCR and acknowledgeable indicators were listed in Appendix A.

## 4. Discussion

Prostate cancer is characterized by its notable heterogeneity followed by a wide variation of oncologic prognosis. The majority of prostate cancers are indolent, while the rest could be very aggressive and even life threatening. Thus, developing high discriminative prognostic models to distinguish low risk of BCR patients from high-risk ones and to provide directly instructive therapeutic assistant is of great importance. In this study, we developed a survival model to predict biochemical recurrence after radical prostatectomy. This model combined conventional radiomic technique for feature extraction and deep learning algorithm for survival analysis. In multicenter validation, the deep radiomic model (DRS-BCR) outperformed three popular clinical conventional models in prediction of BCR-free survival.

Clinicians have made great efforts to identify clinical variable combinations in pursuit of better oncological prediction and controls. Some models have been widely adopted and validated in western countries such as the NCCN stratification, CAPRA/-S score and Gleason grade group system [5]. However, the performance of these models in Asian population have been reported to be suboptimal than those in the European and American population [20]. American-Asian men were reported to be more likely to have unfavorable risk profile with worse prognosis [21]. In the current study, nearly two third patients were classified into intermediate- or high-risk groups, which is in line with Korean and Japanese reports [22,23]. Conversely, most patients in European were low risk [24]. These facts have increasingly drawn our attention for a hypothesis that the entity of prostate cancer in east and west are taking distinct evolution paths. Recently, evidence supporting such differences has been discovered in genetic level. In a large national prostatic genomic analysis, Chinese population were reported to have distinct driver gene profile comparing to American population [25]. Thus, conventional predictive models might not be suitable for Asian people. These facts motivated us to develop a new stratification system according to our own clinical data. In the current study, we developed and validated a deep learning booted radiomic signature (DRS-BCR) to predict BCR-free survival. The model achieved very good performance in all centers, outweighing other four clinical models (NCCN, CAPRA/-S and Gleason grade group system) in the same setting. This was in line with several reports that radiomic models have better performance than conventional ones [13]. 

Prostate cancer is a multifocal malignancy with highly pronounced heterogeneity, treated by multimodal approaches. Patients are generally stratified into different risk groups followed by distinct therapies. Thus, accurate stratification is critical to certain group of patients, especially for high-risk ones. Hormonal therapy is one of the key to prostate cancer. Available evidence indicated that the use of neoadjuvant ADT in RP candidates could reduce positive surgical margin rates, EPE rates and lymph node involvements [26,27]. Of note, these benefits only occurred in high-risk patients [28]. With the widespread application of PSMA-PET-CT [29,30,31] and single-cell sequencing [32,33], early micrometastasis is considered to be one critical cause for biochemical recurrence or progressive metastasis after radical prostatectomy. Therefore, accurate prediction of the risk of biochemical recurrence will help the early use of systemic therapy for the control of systemic micrometastasis in order to obtain longer-term survival benefits. Based on results of predictive models, many ongoing clinical studies were designed to provide adjuvant new-generation antiandrogens for high-risk patients, such as abiraterone (NCT04513717), darolutamide (NCT04484818) and apalutamide (NCT03767244). Compared with the traditional predictive tools, our model can use prebiopsy MR make recurrence risk predictions, so it can be a good basis for stratification to guide the early application of postoperative systemic adjuvant therapy, and even guide the development of neoadjuvant systemic treatment.

Considering the clinical implementation of the DRS-BCR model, we made a comparison with CAPRA-S score by fourfold table (Appendix A). According to the CAPRA-S system, 355 (73.20%) patients were classified into high-risk group, while 260 of them had not developed BCR. Within the 260 patients, only those who harbored latent metastatic cancer cells but had not developed BCR yet could benefit from adjuvant therapies, while the rest “majority” of them were more likely to only suffer from side effects of over treatments. As comparison, DRS-BCR model significantly reduced the proportion of the false positive patients and maintained equivalent high level of negative predictive value (89.4% vs. 86.20%). This indicated that DRS-BCR model can help prevent more than one third patients from overtreatments and keep the undertreatment rate at a similarly low level as CAPRA system.

Comparing with machine learning-based Cox proportional-hazards regression model (CPH), deep survival neural network is able to describe a non-linear relationship between features and survival events. Usually, the relationship between features and survival events is often non-linear [18], which makes conventional linear models (e.g., CPH) difficult to obtain optimal performance. In the current study, imaging features were extracted and primarily selected by conventional radiomic approach. And then, a deep survival neural network algorithm was applied to develop a more complex non-linear model, naming DRS-BCR. Besides, to further mitigating overfitting, strategies such as dropout [34] Early stopping [16], transfer learning [35] were also used.

Additionally, we attempted to incorporate perioperative parameters with DRS-BCR to make a new model (DRC-BCR) in pursuit of better performance. Unluckily, DRC-BCR only demonstrated equivalent performance when comparing with DRS-BCR. This was not in line with those of most previous studies, wherein the final performance of the model can always be improved by incorporating clinical factors [11,36]. It could be assumed that isolated clinical variables can only reflect limited tumor features from one single perspective, as with one color from a prism. With a noninvasive way to analysis all information from a region, it is quite possible for radiomics to provide more complex and deeper level of information to better characterize tumor nature.

Finally, relatively short follow-up is a major limitation of our study. Selecting 3-year and 5-year BCR-free survival in a small cohort decreased the diagnostic power of the model. Further, patients in clinical high-risk group might have received long-term adjuvant ADT or adjuvant RT, which may lead to delayed BCR occurrence. Thus, 10-year long-term follow-up might better reflect the true biological process. Additionally, incorporating of DWI or ADC maps may further improve the overall accuracy and consistency of the model in the future.

## 5. Conclusions

In conclusion, quantitative features of MRI of prostate cancer showed the potential to improve the description of tumor heterogeneity. These features were empowered by deep learning-based method to build a powerful prognostic prediction model for BCR after RP. With the use of this model, it is promising to optimize neoadjuvant or adjuvant systemic therapies for suitable patients.

## Figures and Tables

**Figure 1 cancers-13-03098-f001:**
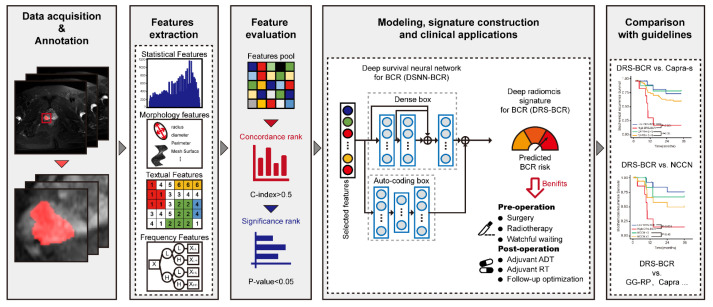
The workflow of radiomics-based model for data preparation, modeling, signature construction, predicting BCR-free survival, and clinical applications. DSNN: deep survival neural network; BCR: biochemical recurrence; DRS: deep radiomic signature; CAPRA: Cancer of Prostate Risk Assessment; NCCN: National Comprehensive Cancer Network; GG: Gleason grade group system.

**Figure 2 cancers-13-03098-f002:**
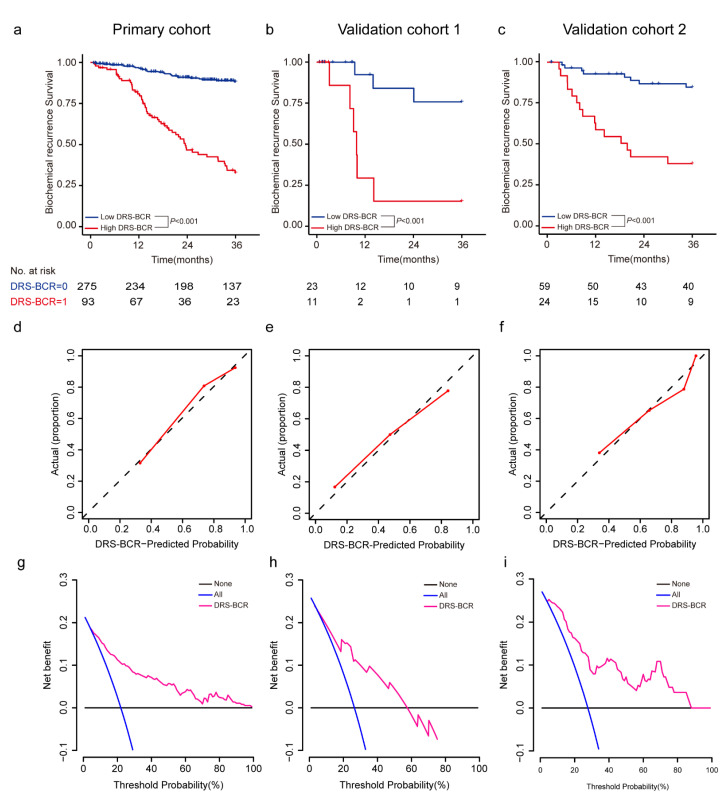
Performance of DRS-BCR for predicting 3-year BCR-free survival. (**a**–**c**) The Kaplan–Meier (K–M) curves of DRS-BCR-free survival for BCR-free survival within three years in the primary cohort, validation cohort 1, and validation cohort 2, respectively. The *p* values were calculated by log-rank test between subgroup with high-risk and low-risk, and significant discrimination was revealed by *p* values less than 0.05. (**d**–**f**) The calibration curves for 3-year BCR. (**g**–**i**) Clinical benefits by decision curve analysis for 3-year BCR. DRS: deep radiomic signature; BCR: biochemical recurrence.

**Figure 3 cancers-13-03098-f003:**
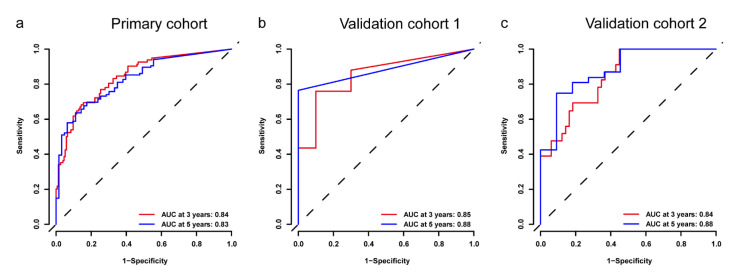
The time-dependence receiver operating characteristic curve (ROC) of DRS-BCR in the primary cohort, validation cohort 1, and validation cohort 2. DRS: deep radiomic signature; AUC: area under the curve.

**Table 1 cancers-13-03098-t001:** Performance of clinical models and deep radiomic models of BCR prediction by univariate cox proportional hazards regression.

	PC (PUTH)	VC (BJFH)	VC2 (PUPH)	*p*
	*n* = 368	*n* = 34	*n* = 83	
No. of BCR event (%)	99 (27.1)	16 (47.1)	31 (37.3)	0.016
Age (Median)	70	68.5	68.0	0.275
PSA (Median)	10.7	9.89	13.4	0.883
GG-NB				0.125
1	138	13	29	
2	114	10	23	
3	63	6	14	
4	89	3	11	
5	81	2	6	
cT				0.436
2	221	15	44	
3	264	19	39	
4	9			
pT				0.687
1	2		1	
2	303	22	57	
3	165	12	22	
4	15		3	
GG-RP				0.173
1	82	4	13	
2	127	10	28	
3	93	8	18	
4	65	9	8	
5	118	3	16	
SM				0.172
Positive	170	12	38	
Negative	315	21	45	
EPE				0.639
Positive	171	11	25	
Negative	314	23	58	
SVI				0.940
Positive	67	4	11	
Negative	418	30	72	
CAPRA				0.248
Low	53	9	10	
Intermediate	168	12	34	
High	148	13	39	

Note: BCR: biochemical recurrence; PC, primary cohort; VC, validation cohort; PUTH, Peking University Third Hospital; BJFH, Beijing Friendship Hospital; PUPH, Peking University People’s Hospital; PSA, prostate specific antigen; GG-NB, Gleason grade group of needle biopsy; cT. clinical T stage; pT, pathological T stage; GG-RP, Gleason grade group of radical prostatectomy; SM, surgical margin; EPE, extracapsular extension; SVI, seminal vesicle invasion; CAPRA, Cancer of Prostate Risk Assessment Score.

**Table 2 cancers-13-03098-t002:** Patient characteristics.

	HR	*p*	PC (*n* = 369)	VC1 (*n* = 34)	VC2 (*n* = 83)
GG-RP	1.645	0.001	0.583 [0.53–0.636]	0.564 [0.419–0.709]	0.689 [0.601–0.777]
CAPRA-S	1.339	<0.001	0.677 [0.62–0.734]	0.654 [0.49–0.818]	0.654 [0.544–0.764]
NCCN	1.9022	<0.001	0.586 [0.548–0.624]	0.535 [0.408–0.662]	0.583 [0.498–0.668]
CAPRA	1.306	<0.001	0.677 [0.618–0.736]	0.552 [0.385–0.719]	0.614 [0.509–0.719]
DRS-BCR	1.705	<0.001	0.802	0.811	0.794
			[0.758–0.846]	[0.722–0.9]	[0.718–0.87]
DRC-BCR	1.654	<0.001	0.807 [0.76–0.854]	0.794 [0.685–0.903]	0.8 [0.723–0.877]

Note: GG-RP: Gleason grade group of radical prostatectomy; BCR: biochemical recurrence; PC, primary cohort; VC, validation cohort; HR: hazard ratio; CAPRA: Cancer of Prostate Risk Assessment; NCCN: National Comprehensive Cancer Network; DRS-BCR: deep radiomic signature of biochemical recurrence; DRC-BCR: deep radiomic combing signature of biochemical recurrence. *p* values were two-sided.

## Data Availability

The related data and materials are available for sharing upon request to Jian Lu and Jie Tian.

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
