# Peer review of "Deep Learning with Quantitative Features of Magnetic Resonance Images to Predict Biochemical Recurrence of Radical Prostatectomy: A Multi-Center Study"

_cancers, 2021, doi:10.3390/cancers13123098_

Round 1

Reviewer 1 Report

This manuscript focused on an important clinical issue of the prediction of biochemical recurrence after radical prostatectomy and developed an interesting algorithm by combining radiomics and deep learning algorithms. In particular, the methods were developed and validated based on multi-center data with a large population. In general, the paper is well written. It would be nice if the authors can further improve the following points: 1. It is still debating about the choice of either radiomics or deep learning in the community. In general, I like the combination of radiomics and deep learning. However, it would be nice if the authors can clarify the rationals behind that. Theortically deep learning can directly extract features from the data. The integration of artificially designed radiomics features may have complementary value. But an in-depth insight would be helpful. 2. Multi-parametric imaging is routinely applied in urological diagnosis. It is not clear to me about the rationals of only using T2w without DWI or ADC? 3. More evidence demonstrated that adjuvant radiotherapy is only recommended in high-risk patients after radical prostatectomy and is gradually being replaced by early salvage radiotherapy (GETUG). More recently, adjuvant ADT is only applied in pathological N-positive patients after radical prostatectomy. It would be nice if the authors can perform a subgroup analysis in N+/- patients. 4. In Table 1, Gleason Score(GS-NB) and Grade Group(GG-RP)were used simultaneously, which is not clear for data stratification. Furthermore, a lot of abbreviations were not defined in tables and supplementary tables. 5. The authors mentioned the 10-year biochemical recurrence rate after radical prostatectomy can be up to 27%. However, the actual 3-year biochemical recurrence rates in the three cohorts were 27%, 47%, and 37%. It would be nice if the authors can clarify the reason for the higher biochemical recurrence rate in a much shorter period in all three centers. 6. Some deep learning studies have revealed the prognosis value of FDG PET or PSMA PET in the prostate or other cancers, it would be nice if the authors can also mention that. 7. Micrometastasis may be a critical problem for recurrence and PSMA-guided radioligand therapy is more and more applied in the treatment of metastatic prostate cancer. I am not sure if PSMA-guided radioligand therapy might be an option as adjuvant radiotherapy in the future?

Author Response

Point-by-point responses

Title: Deep learning with quantitative features of magnetic resonance images to predict biochemical recurrence of Radical prostatectomy: A multi-center study

Dear Editors and Reviewers:

We appreciate you very much for your constructive comments and suggestions on our manuscript. We have studied your comments carefully and revised the manuscript following your suggestions. The reviewers’ concerns or suggestions were also included in this reply for convenience.

Replies to editors’ and reviewers’ comments are listed as follows. The comments of the editors and reviewers are in italic, and our responses following each comment are in regular font. The revised contents in the manuscript are shown in yellow background.

Sincerely,

Ye Yan, Lizhi Shao, Zhenyu Liu, Wei He, Guanyu Yang, Jiangang Liu, Haizhui Xia, Yuting Zhang, Huiying Chen, Cheng Liu, Min Lu, Lulin Ma, Kai Sun, Xuezhi Zhou, Xiongjun Ye, Lei Wang, Jie Tian and Jian Lu

Reviewer 1:

  1. It is still debating about the choice of either radiomics or deep learning in the community. In general, I like the combination of radiomics and deep learning. However, it would be nice if the authors can clarify the rationals behind that. Theortically deep learning can directly extract features from the data. The integration of artificially designed radiomics features may have complementary value. But an in-depth insight would be helpful.

[Response] Deep learning is a class of machine learning algorithms that[1] uses multiple layers to progressively extract higher-level features from the raw input[2]. Learning can be supervised, semi-supervised or unsupervised. Deep-Learning includes deep neural networks, deep belief networks, recurrent neural networks, and convolutional neural networks[3]. The adjective "deep" in deep learning refers to the use of multiple layers in the network. In our study, we employed a new multi-layer neural network based on the idea of ResNet[4], massive, quantified image features and Loss function for survival problem designed for a neural network for a prognosis prediction model. Whether it is from feature flux, model structure, or the process of constructing a model, it is in line with the definition of deep learning by artificial intelligence scientists. Therefore, we carefully adopted Deep learning as the title in the article.

Theoretically, deep learning can directly extract features from the data, but usually, it relies on large-scale data samples for a satisfying performance, especially in medical image analysis[5], which is the basic requirement of data-driven methods. Radiomics is defined as a specialist-driven method for modeling, which part of the dependence on data is alleviated by the prior knowledge of experts becoming a consensus. Some research results have shown that the combination of expert's prior knowledge and deep learning can help the model converge faster and obtain better model performance[6]. At the same time, prostate cancer has occult and multifocal characteristics in imaging examination, and the use of deep learning methods to guide the attention area still requires large-scale data as support. Therefore, a combination of both specialist-driven and data-driven methods is an effective way for a better model under the limitation of data.

  1. Multi-parametric imaging is routinely applied in urological diagnosis. It is not clear to me about the rationals of only using T2w without DWI or ADC?

[Response] Based on the consensus of radiologists and clinicians, T2w is critical for the diagnosis of prostate cancer. T2W has high tissue resolution and can provide good anatomical details. Especially when DWI and DCE are inadequate, T2W has higher diagnostic power. Limited by scanning protocol deviations (from 3 institutions) and our data contains insufficient DWI sequences (poor resolution and inadequate B value), which is the core reason why we used T2W.

  Thank you for your reminder, actually, on one hand, we are collecting new data with more consistent DWI sequences with better resolution for future work (follow-up information is maturing), and on the other hand, we are preparing to initiate a prospective observational clinical trial to validate the model, we are waiting for ethnic’s approval currently.

  1. More evidence demonstrated that adjuvant radiotherapy is only recommended in high-risk patients after radical prostatectomy and is gradually being replaced by early salvage radiotherapy (GETUG). More recently, adjuvant ADT is only applied in pathological N-positive patients after radical prostatectomy. It would be nice if the authors can perform a subgroup analysis in N+/- patients.

[Response] Thank you for your advice. We are longing to performed detailed analysis of pN status and we have documented pN information as well. However, after careful thinking and discussion, we finally gave up subgroup analyzing for lymph nodes status. There are three crucial reasons that influenced the above decision. First, not all patients have had lymph node dissections. Second, the reports of positive lymph nodes after surgery were not documented according to the anatomical sections, so we cannot confirm whether the lymph nodes that were suspicious positive on MR correspond to the actual positive ones. Third, because the lymph node dissection rate varies greatly among different surgeons, and there were non-negligible false negative rates in the diagnosis of preoperative lymph node status, the false negative rate reported in the final report will objectively affect the reliability of modeling.

  1. In Table 1, Gleason Score(GS-NB) and Grade GroupGG-RPwere used simultaneously, which is not clear for data stratification. Furthermore, a lot of abbreviations were not defined in tables and supplementary tables.

[Response] Thank you for your suggestions. We have transferred GS into GG form in all tables and the main part of the manuscript. And we checked all abbreviations and defined each one in all tables. We sincerely apologize for our carelessness that rendered your reading experiences.

  1. The authors mentioned the 10-year biochemical recurrence rate after radical prostatectomy can be up to 27%. However, the actual 3-year biochemical recurrence rates in the three cohorts were 27%, 47%, and 37%. It would be nice if the authors can clarify the reason for the higher biochemical recurrence rate in a much shorter period in all three centers.

[Response] Thank you for pointing out this for us. The prostate cancer populations in Europe/America and Asia are very different. Recent studies have shown that there are huge phenotypic differences between European/American and Chinese prostate cancer patients at the driver gene level, and the accumulation of discrepancies at the transcriptome and protein levels ultimately leads to very different tumor characteristics and epidemiological differences. Most European and the American prostate cancer patients at first diagnosis tend to be low risk, while the proportion of high-risk prostate cancer patients in Asia is significantly higher, with more rapid progress and worse prognosis. Of the three cohort in this study, the majority of patients enrolled were classified in at least intermediate group, which is quite different with reports of western centers. We believe this is the main reason that we tend to have more BCR in short period. Through multiple databases analysis, we have calculated the difference in the proportion of prostate cancer clinical stages in different countries (See table below).

Supplementary Table. Comparison between clinical stages among different countries.

Sweden[7]

America[8]

Korean[9]

Japan[10]

Chinese

N

139515

243433

7608

7768

1766

T1(%)

49.37

45

69.5

33.2

20.3

T2(%)

35.61

44.5

16.3

42

54.4

T3-T4(%)

16.02

10.6

14.2

24.8

25.3

Database

PCBaSe 4.0

SEER

K-CAP

Nara Medical University

Peking University

  1. Some deep learning studies have revealed the prognosis value of FDG PET or PSMA PET in the prostate or other cancers, it would be nice if the authors can also mention that.

[Response] Thank you for your kindly reminder. PSMA-PET has significantly improved the diagnostic power for detecting PCa lesions, especially for micrometastatic small volume lesions. Recently, the combination of deep learning and PSMA/FDG-PET is widely used in the diagnosis, therapeutic evaluation and prediction of prostate cancer, such as nodal staging[11], risk stratification[12], recurrent lesion detection[13] and so on. We have added these into the discussion.

  1. Micrometastasis may be a critical problem for recurrence and PSMA-guided radioligand therapy is more and more applied in the treatment of metastatic prostate cancer. I am not sure if PSMA-guided radioligand therapy might be an option as adjuvant radiotherapy in the future?

[Response] Thank you for your inspiring advice. In the past two years, several large phase III RCTs (GETUG, RAVES, RADICALS-RT) have eventually shown that adjuvant radiotherapy does not show a survival advantage compared with early rescue radiotherapy. Conversely, early systemic micrometastasis of prostate cancer might play a key role in oncological control, which negatively makes adjuvant RT and early salvage RT equal. Thus, more efficient systemic therapies are needed such as second-generation antiandrogen treatment and PSMA-labeled radioligand therapies.

Recently, PSMA-targeted radioligand therapies have been generally approved for mCRPC[14], while some attempts have also been made in testing efficacies of these therapies in early stage of PCa[15]. PSMA is highly sufficient in detecting and targeting metastasis with very small volume. It could be assumed that adjuvant use of PSMA-labeled agents in high-risk patients could benefit long-term survival. BCR is a surrogate for early recurrence or metastasis, our model significantly enables urologists to identify patients with high risk of recurrence (or micrometastasis) after RP, which might facilitated future adjuvant radiotherapies.

Reference

  1. LeCun Y, Bengio Y, Hinton G: Deep learning. Nature 2015, 521(7553):436-444.
  2. Hu JY, Niu HL, Carrasco J, Lennox B, Arvin F: Voronoi-Based Multi-Robot Autonomous Exploration in Unknown Environments via Deep Reinforcement Learning. Ieee T Veh Technol 2020, 69(12):14413-14423.
  3. Deng L, Yu D: Deep learning: methods and applications. Foundations and trends in signal processing 2014, 7(3–4):197-387.
  4. Wu Z, Shen C, Van Den Hengel A: Wider or deeper: Revisiting the resnet model for visual recognition. Pattern Recognition 2019, 90:119-133.
  5. Shen D, Wu G, Suk H-I: Deep learning in medical image analysis. Annual review of biomedical engineering 2017, 19:221-248.
  6. Greenspan H, Van Ginneken B, Summers RM: Guest editorial deep learning in medical imaging: Overview and future promise of an exciting new technique. IEEE Transactions on Medical Imaging 2016, 35(5):1153-1159.
  7. Zelic R, Garmo H, Zugna D, Stattin P, Richiardi L, Akre O, Pettersson A: Predicting Prostate Cancer Death with Different Pretreatment Risk Stratification Tools: A Head-to-head Comparison in a Nationwide Cohort Study. Eur Urol 2020, 77(2):180-188.
  8. Dess RT, Hartman HE, Mahal BA, Soni PD, Jackson WC, Cooperberg MR, Amling CL, Aronson WJ, Kane CJ, Terris MK et al: Association of Black Race With Prostate Cancer-Specific and Other-Cause Mortality. JAMA Oncol 2019, 5(7):975-983.
  9. Ahn H, Kim HJ, Jeon SS, Kwak C, Sung GT, Kwon TG, Park JY, Paick SH: Establishment of Korean prostate cancer database by the Korean Urological Oncology Society. Investig Clin Urol 2017, 58(6):434-439.
  10. Tanaka N, Nakai Y, Miyake M, Anai S, Inoue T, Fujii T, Konishi N, Fujimoto K: Trends in risk classification and primary therapy of Japanese patients with prostate cancer in Nara urological research and treatment group (NURTG) - comparison between 2004-2006, 2007-2009, and 2010-2012. BMC Cancer 2017, 17(1):616.
  11. Hartenstein A, Lübbe F, Baur AD, Rudolph MM, Furth C, Brenner W, Amthauer H, Hamm B, Makowski M, Penzkofer T: Prostate cancer nodal staging: using deep learning to predict 68 Ga-PSMA-positivity from CT imaging alone. Scientific reports 2020, 10(1):1-11.
  12. Cysouw MC, Jansen BH, van de Brug T, Oprea-Lager DE, Pfaehler E, de Vries BM, van Moorselaar RJ, Hoekstra OS, Vis AN, Boellaard R: Machine learning-based analysis of [18 F] DCFPyL PET radiomics for risk stratification in primary prostate cancer. European journal of nuclear medicine and molecular imaging 2021, 48(2):340-349.
  13. Lee JJ, Yang H, Franc BL, Iagaru A, Davidzon GA: Deep learning detection of prostate cancer recurrence with 18 F-FACBC (fluciclovine, Axumin®) positron emission tomography. European journal of nuclear medicine and molecular imaging 2020, 47(13):2992-2997.
  14. Hofman MS, Emmett L, Sandhu S, Iravani A, Joshua AM, Goh JC, Pattison DA, Tan TH, Kirkwood ID, Ng S et al: [(177)Lu]Lu-PSMA-617 versus cabazitaxel in patients with metastatic castration-resistant prostate cancer (TheraP): a randomised, open-label, phase 2 trial. Lancet 2021, 397(10276):797-804.
  15. Prive BM, Peters SMB, Muselaers CHJ, van Oort IM, Janssen MJR, Sedelaar JPM, Konijnenberg MW, Zamecnik P, Uijen MJM, Schilham MGM et al: Lutetium-177-PSMA-617 in Low-Volume Hormone-Sensitive Metastatic Prostate Cancer: A Prospective Pilot Study. Clin Cancer Res 2021.

Reviewer 2 Report

In this paper, the authors retrospectively reviewed clinical and MRI imaging information on 485 patients who underwent radical prostatectomy at 3 institutions in China to create a deep learning based model to predict BCR. They found that their model had a C index of 0.082, which was higher than conventional models (CAPRA, NCCN risk groups and Gleason grade).

They did find good results for what they were looking for but the presentation is not very clear and I have some comments.

  • Significant grammar and typo editing is needed for this to be published in an English language journal.
  • Intro
    • Line 48: Wound not say that radical prostatectomy is the mainstay of treatment for localized prostate cancer. Surgery is one option along with XRT.
    • Line 53: I would not say that guidelines recommend adjuvant ADT to reduce BCR in any scenario. First, the goal of post op XRT is to reduce metastases and improve survival, not target BCR. Also, recent trial data confirms no benefit to adjuvant over salvage ADT (RADICALS). You need to reframe a lot of your writing to fit the most up to date data .
  • Methods:
    • Be more explicit with your definitions in the inclusion criteria. Stage? You included cT4 patients?
    • Table 1 should be in the results, not methods section
    • Abbreviations should be in the text in addition to the table captions. It is not easy to read the text and have to refer to the tables for abbreviations.
    • Would explain Figure 1 in the text better.
  • Results
    • Too much data in the supplementary tables
    • You don’t tell us what variables are included in DRC-BCR
    • The tables can’t stand on their own. Need more descriptive titles
  • Discussion
    • Again, what’s the clinical benefit? Need to mention updated data on adjuvant vs salvage treatment
    • Did you test a simpler model with PIRADS score and clinical variables?
  • Conclusions
    • Don’t fit with the test of the paper

Author Response

Point-by-point responses

Title: Deep learning with quantitative features of magnetic resonance images to predict biochemical recurrence of Radical prostatectomy: A multi-center study

Dear Editors and Reviewers:

We appreciate you very much for your constructive comments and suggestions on our manuscript. We have studied your comments carefully and revised the manuscript following your suggestions. The reviewers’ concerns or suggestions were also included in this reply for convenience.

Replies to editors’ and reviewers’ comments are listed as follows. The comments of the editors and reviewers are in italic, and our responses following each comment are in regular font. The revised contents in the manuscript are shown in yellow background.

Sincerely,

Ye Yan, Lizhi Shao, Zhenyu Liu, Wei He, Guanyu Yang, Jiangang Liu, Haizhui Xia, Yuting Zhang, Huiying Chen, Cheng Liu, Min Lu, Lulin Ma, Kai Sun, Xuezhi Zhou, Xiongjun Ye, Lei Wang, Jie Tian and Jian Lu

Reviewer 2

1.Significant grammar and typo editing is needed for this to be published in an English language journal.

[Response] We invited colleagues who are native English speakers to help us check the manuscript and correct the errors

  1. Line 48: Wound not say that radical prostatectomy is the mainstay of treatment for localized prostate cancer. Surgery is one option along with XRT.

[Response] Thank you for your kindly reminder, we have altered this sentence to “Radical prostatectomy (RP) is one option of the multimodal approaches for organ-confined or locally advanced PCa”.

  1. Line 53: I would not say that guidelines recommend adjuvant ADT to reduce BCR in any scenario. First, the goal of post op XRT is to reduce metastases and improve survival, not target BCR. Also, recent trial data confirms no benefit to adjuvant over salvage ADT (RADICALS). You need to reframe a lot of your writing to fit the most up to date data .

[Response] We are really grateful to you for pointing out this problem for us. We started this project from Q1 2018. Because radiotherapy is one of the most important treatments after RP, our original intention of designing this model is to predict the risk of biochemical recurrence at early phase, so as to more accurately identify the population who can be benefit from early adjuvant radiotherapy. However, according to your reminder, three large scale RCTs (RAVES, RADICALS-RT and GETUG AFU 17) demonstrated negative incremental OS benefit of adjuvant RT comparing to early salvage RT at the time of BCR[1-3]. Results of these trials made the model we originally designed greatly compromised for guiding adjuvant RT.

Single cell sequencing revealed the existence of early micro-metastases of PSA positive PCa cells[4], and BCR is not only a landmark of local recurrence but more importantly an indicator of early micro-metastases[5]. This might offer an explanation why adjuvant RT and early salvage RT have equivalent impact on OS, because hidden systemic micro-metastases require more intensive systemic therapy rather than local control plus only ADT. Thus, several ongoing pipeline strategies focusing second-generation antiandrogen therapies in post-operative high-risk patients are being carried out (See table below, data from ClinicalTrials.gov). In this scenario, predicting BCR at early phase of the disease might enable oncologist to identify patients of high probabilities for biochemical recurrence. This, in turn, has important implications for post-operative therapeutic selection and patient counseling. We reframed our writing (Introduction, Discussion and Conclusion) in accordance with latest data to be more explicit in clinical implication and potential benefit.

Name or NCT No.

Intervention

NCT00116142; P.C. 6/20

(RT + ADT ± docetaxel)

ATLAS P.C. 12/22

(ADT+ RT ± APA)

PrTK03 P.C. 6/23

(RT ± ProstAtak/GMCI/CAN-2409 ± ADT)

ENZARAD; P.C. 12/23

(ENZA vs. NSAA w/ RT+ADT)

PROTEUS; P.C. 4/24

(RP w/ neo/adj APA + ADT vs. Pbo + ADT)

NCT03777982; P.C. 12/26

(If PSA >0.1ng add APA + AA to ADT vs. ADT)

DASL-HiCaP; P.C. 1/28

(ADT + RT ± DARO)

NCT04484818; P.C. 5/28

(Adj. DARO + ADT vs. pbo + ADT; Decipher high)

PREDICT-RT; P.C. 12/33

(RT + ADT ± APA/AA based on Decipher score)

  1. Be more explicit with your definitions in the inclusion criteria. Stage? You included cT4 patients?

[Response] Thank you for your reminder. According to our study design, cT4 is not an exclusive criterion, in this study, 9 cases of cT4 was enrolled, and a total of 18 patients were evaluated as pT4. We carefully rechecked these cases, all 18 cases were microscopically confirmed with bladder wall invasions, and all had negative margin with PSA nadir < 0.1 ng/ml within 3 weeks postoperatively before adjuvant ADT implementation.

  1. Table 1 should be in the results, not methods section

[Response] Thank you for your advice, we have moved Table 1 to the Result session.

  1. Abbreviations should be in the text in addition to the table captions. It is not easy to read the text and have to refer to the tables for abbreviations.

[Response] Thank you for your reminder. We rechecked the abbreviations in the text and table captions. We have made a complete description of the abbreviations in the text for the first time, and added explanations at the end of the table for the abbreviations involved in the table for a clearer statement.

  1. Would explain Figure 1 in the text better.

[Response] We added introduction of Figure 1 in the text according to your reminder. (Page 4, line 116-121)

“The first step included data acuqisition of images and tumor sementation. Then, we extracted radiomic features from tumor area for tumor heterogeneity. Thirdly, feature evalutaion based on rank is used to roughly remove features that are of little value in modeling based on statistical approaaches. Finally, prognosis prediction of BCR was constructed and valudated in our multi-center dataset. Comparsion with guidelines was also supplemented.”

  1. Too much data in the supplementary tables

[Response] We have done a lot of detailed work in the research process for a better rearch, and hope to share it with all readers, which hope to have the opportunity to help them with related research. Due to the length of the journal, after repeated choices, we finally determined that the most important contribution of our work should be placed in the main text. Other meaningful content will be open throughsupplementary materials online. Thank you very much for your understanding.

  1. You don’t tell us what variables are included in DRC-BCR

[Response] The deep radiomic signature for BCR (DRC-BCR) is based on radiomic features from tumor area of T2WI after feature evaluation. We summarized the modeling features and put them in a separate .xlsx file. We have supplemented this part of the content in the method. (Page 5, line 152-165)

  1. The tables can’t stand on their own. Need more descriptive titles

[Response] Thank you for your advice. We adopted your suggestions and made a modification to our table titles to be more explicit to read.

  1. Again, what’s the clinical benefit? Need to mention updated data on adjuvant vs salvage treatment

[Response] Thank you again, for your generous comment on this issue. We totally accept your advice and made a thorough statement within the response to your second comment. We hope our new writing can fit your idea and thank you for your understanding. Your comment really means too much for us.

  1. Did you test a simpler model with PIRADS score and clinical variables?

[Response] Firstly, PIRADS is used for distinguishing cancer lesions from benign lesions. Compared with pathological evaluation, especially postoperative pathological evaluation, its prognostic ability is limited. Our data were collected over 9 years with two major version of PIRADS system and several minor modifications on scanning protocols, which made our imaging data difficult to be evaluated according to any single version of PIRADS. Secondly, we actually did re-evaluate all cases according to PIRADS V2.1, limited by inadequate scanning sequence, cases of early years are incomparable with those of recent years. Considering the above two reasons, we decided to abandon PIRADS for modeling.

However, we have organized clinical variables with statistical significance based on available clinical data for a BCR prediction model, and evaluate them in the Table S2-5. To assess the independent prognostic power of each clinical parameter, we grouped them as preoperative clinical parameters (PSA, cT stage, GS-NB, PPB) and postoperative parameters (GG-RP, surgical margin, extracapsular extension, seminal vesicle invasion) in a univariable cox regression model. The parameters were normalized by CAPRA (cT stage, GS-NB, PPB) or CAPRA-S (PSA, GG-RP, surgical margin, extracapsular extension, seminal vesicle) criteria. The results of the univariate analysis are shown in Table S2 and subsection of Patient characteristics in results. (Page 183-200)

“We integrated all significant clinical variables (P < 0.05), and grouped them by the acquisition time (pre-operative, post-operative, a combination of both pre- and post-operative), and built three clinical prognostic models for BCR prediction, respectively (Table S3, S4, S5). We termed the clinical variables as clinical signature (CS). The combi-nation of pre- and post-operative clinical variables (CS-combine) yielded C-index [95% CI] of 0.693 [0.634-0.752] in PC, 0.651 [0.471-0.831] in VC1 and 0.641 [0.529-0.753] in VC2, which is the highest among the three.”

  1. Don’t fit with the test of the paper.

[Response] Thanks again for your kindly reminder. We have rephrased the clinical benefit and reoriented the conclusion in accordance with updated data.

Reference

  1. Kneebone A, Fraser-Browne C, Duchesne GM, Fisher R, Frydenberg M, Herschtal A, Williams SG, Brown C, Delprado W, Haworth A et al: Adjuvant radiotherapy versus early salvage radiotherapy following radical prostatectomy (TROG 08.03/ANZUP RAVES): a randomised, controlled, phase 3, non-inferiority trial. Lancet Oncol 2020, 21(10):1331-1340.
  2. Parker CC, Clarke NW, Cook AD, Kynaston HG, Petersen PM, Catton C, Cross W, Logue J, Parulekar W, Payne H et al: Timing of radiotherapy after radical prostatectomy (RADICALS-RT): a randomised, controlled phase 3 trial. Lancet 2020, 396(10260):1413-1421.
  3. Sargos P, Chabaud S, Latorzeff I, Magne N, Benyoucef A, Supiot S, Pasquier D, Abdiche MS, Gilliot O, Graff-Cailleaud P et al: Adjuvant radiotherapy versus early salvage radiotherapy plus short-term androgen deprivation therapy in men with localised prostate cancer after radical prostatectomy (GETUG-AFU 17): a randomised, phase 3 trial. Lancet Oncol 2020, 21(10):1341-1352.
  4. Chen S, Zhu G, Yang Y, Wang F, Xiao YT, Zhang N, Bian X, Zhu Y, Yu Y, Liu F et al: Single-cell analysis reveals transcriptomic remodellings in distinct cell types that contribute to human prostate cancer progression. Nat Cell Biol 2021, 23(1):87-98.
  5. Murray NP, Aedo S, Fuentealba C, Reyes E, Salazar A, Lopez MA, Minzer S, Orrego S, Guzman E: Minimal Residual Disease Defines the Risk and Time to Biochemical Failure in Patients with Pt2 and Pt3a Prostate Cancer Treated With Radical Prostatectomy: An Observational Prospective Study. Urol J 2020, 17(3):262-270.

Round 2

Reviewer 1 Report

The authors properly addressed my comments, which is highly appreciated. I don't have further comments.